# Learning Reward Functions from Scale Feedback

**Nils Wilde**[*†]     **Erdem Bıyık**[*‡]     **Dorsa Sadigh**[‡§]     **Stephen L. Smith**[†]

[†] Electrical and Computer Engineering, University of Waterloo
[‡] Electrical Engineering, Stanford University
[§] Computer Science, Stanford University
{nwilde,stephen.smith}@uwaterloo.ca, {ebiyik,dorsa}@stanford.edu
[*] Authors contributed equally.

**Abstract:** Today's robots are increasingly interacting with people and need to efficiently learn inexperienced user's preferences. A common framework is to iteratively query the user about which of two presented robot trajectories they prefer. While this minimizes the users effort, a strict choice does not yield any information on *how much* one trajectory is preferred. We propose scale feedback, where the user utilizes a slider to give more nuanced information. We introduce a probabilistic model on how users would provide feedback and derive a learning framework for the robot. We demonstrate the performance benefit of slider feedback in simulations, and validate our approach in two user studies suggesting that scale feedback enables more effective learning in practice.

**Keywords:** HRI, reward learning, learning from choice, active learning

## 1 Introduction

While autonomous robots are able to accomplish an increasing variety of tasks, a key challenge that still remains is *how* they should pursue and trade off between their goals. In recent years, there has been significant work on interactively learning user preferences of robot behaviors [1–15]. Usually, the user is provided with one or more robot trajectories, and is asked to provide feedback through pairwise comparisons [2–8], rankings of the trajectories [16, 17], or physical feedback [18, 12, 14]. The underlying reward function governing human preferences can then be learned through this implicit feedback. Specifically, one framework with minimal complexity for the user is *learning from choice feedback* [2–8], where the robot demonstrates two alternative trajectories for some task. The user then simply chooses their preferred behavior allowing the robot to infer an underlying reward function for the user preferences.

Choice feedback, although simple to collect, is limiting in a number of ways. Consider the example shown in Fig. 1, where a robot is tasked to serve a drink to a customer. The customer might have different preferences over the type of drink to have (milk, orange juice, or water), or the specifics of the trajectory the robot takes (e.g., if it goes over the stove or around it which can affect the temperature of the drink or the likelihood of the robot accidentally hitting the pan handle). A strict choice feedback between two trajectories does not really capture these intricacies of human preferences. We thus need to have a more expressive way of collecting data from humans. Our key insight is that allowing users to provide a scaled approach on a slider (as shown in Fig. 1) can provide a more expressive medium for learning from humans and capture nuances in their preferences.

In this work, we propose *scale feedback* as a new mode of interaction: Instead of a strict question on which of the two proposed trajectories the user prefers, we allow for more nuanced feedback using a slider bar. We design a Gaussian model for how users provide scale feedback, and learn a reward function capturing human preferences. Similar to prior work in robotics, we assume this reward is a linear function of a set of features [19, 11, 13, 7], where the main task of learning from scale feedback is to recover the weights of this reward function. To learn in a data-efficient manner, we actively generate our queries to the user, i.e., pairs of trajectories demonstrated to a user similar to Fig. 1, by optimizing two well-known objectives of information gain [4] and max regret [5].

We demonstrate the performance benefit of scale feedback over choice in a driving simulation. Further, we investigate its practicality in two user studies with the real robot experiment shown in Fig. 1. Our results suggest scale feedback leads to significant improvements in learning performance.

5th Conference on Robot Learning (CoRL 2021), London, UK.

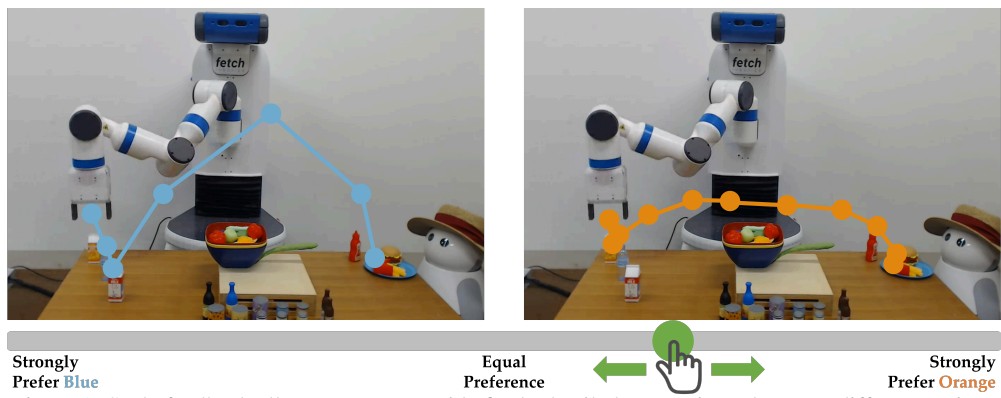

Figure 1: Scale feedback allows users to provide finely detailed comparisons between different options.

## 2 Related Work

Learning from human feedback is an important problem in developing interactive robots that work alongside humans. Researchers study learning from demonstrations [19–21], corrections [22, 23, 14], ordinal feedback [24, 8], rankings [16, 17, 25, 26], critiques [27, 28], and choices [2–5, 29].

While demonstrations usually are very informative, they are not always viable: Demonstrating the desired behavior might require a high level of expertise [30], or can be difficult in high-order systems [31–33]. Choice questions minimize interface complexity and mental effort for the user. However, when the user is indifferent towards both options, learning becomes difficult since users may become noisier in their responses. Thus, [7, 4, 6] investigate modifications of learning from choice where users can also answer *About Equal*. These two forms of choice feedback are usually referred to as *strict* and *soft* choice. When the user chooses the neutral answer, the robot learns to assign about equal reward to the presented trajectories. While choice feedback provides an easy medium for learning from humans, it provides at most one bit of information. Thus, to effectively learn from choice feedback, we often need to actively generate the queries made to humans. Previous work has investigated different auxiliary measures that are greedily optimized to enable efficient learning, including expected volume removal [2], information gain [4], and regret [5].

In the proposed scale feedback framework, we take the soft choice approach one step further: Instead of three discrete values for feedback (prefer A, prefer B, neutral) users give quasi-continuous feedback. This allows the user to indicate by *how much* they prefer one option over the other.

Slider bars have been used in robotics for tuning parameters [34]. More related to our work, Cabi et al. [35] proposed using them for *reward sketching*. Instead of assigning a numerical preference between presented options, users continuously indicate the robot's progress towards some goal. However, this requires users to assign scores to different parts of trajectories. Developing the scale feedback for preference-based learning, we retain the ease of comparing trajectories.

## 3 Problem Formulation

We now introduce the notation we use in this paper and formulate the learning problem.

**Reward function.** We consider the scenario where a robot needs to customize its behavior to the preferences of a user Alice. We assume Alice evaluates robot paths $P \in \mathcal{P}$ based on a vector of features $\boldsymbol{\phi}^P = [\phi_1(P), \ldots, \phi_n(P)]$. Similar to prior works in robotics [19, 11, 13, 7], we define a linear reward function $r$ that assigns a numerical value to a path $P$ by weighting a set of features:

$$r(P, \boldsymbol{w}) = \boldsymbol{\phi}^P \cdot \boldsymbol{w}. \tag{1}$$

These features are usually provided by a domain expert incorporating the core factors that the reward needs to capture, e.g., collision with other objects, or distance to the goal. Further, the robot has access to a motion planner that finds an optimal path given a set of weights, i.e., the planner is a (deterministic) function $\rho : \mathbb{R}^n \to \mathcal{P}$ where $\rho(\boldsymbol{w}) = \arg\max_{P \in \mathcal{P}} r(P, \boldsymbol{w})$.

**Regret.** Similar to [5], we define the *regret* between any two weights $(\boldsymbol{w}, \boldsymbol{w}')$ as the difference in the reward $\boldsymbol{w}'$ assigns to the paths $\rho(\boldsymbol{w})$ and $\rho(\boldsymbol{w}')$:

$$R(\boldsymbol{w}, \boldsymbol{w}') = \boldsymbol{\phi}(\rho(\boldsymbol{w}')) \cdot \boldsymbol{w}' - \boldsymbol{\phi}(\rho(\boldsymbol{w})) \cdot \boldsymbol{w}', \tag{2}$$

which quantifies the suboptimality when the true weights are $\boldsymbol{w}'$, but the path is optimized using $\boldsymbol{w}$.

**Learning.** Let $\boldsymbol{w}^*$ denote Alice's weights for the reward function. These weights are not known to the robot; the only information initially available is a prior distribution $\mathbb{P}(\boldsymbol{w} = \boldsymbol{w}^*)$. The robot

learns $\boldsymbol{w}^*$ by iteratively presenting her with two paths $P$ and $Q$ for $K$ iterations. We extend the *learning from choice* framework, where users simply indicate the path they prefer, to a setting where they instead provide a more finely detailed *scale feedback*.

**Definition 1** (Scale feedback). Presented with two paths $P$ and $Q$, Alice returns numerical feedback $\psi \in [-1, 1]$. If $\psi = 0$, this means Alice has no preference between the paths, $\psi = 1$ equals a strong preference for path $P$ and $\psi = -1$ a strong preference for path $Q$.

From an interface design and expressiveness perspective, it is undesirable to have users give a numerical value for $\psi$. Instead, they can express such a feedback with a slider bar with a more fine-grained set of options. An example is illustrated in Fig. 1. We let $D_K = \{(P_1, Q_1, \psi_1), \ldots, (P_K, Q_K, \psi_K)\}$ be the set of recorded user feedback.

**Performance Measures.** Let $\hat{\boldsymbol{w}}$ be the robot's estimate of $\boldsymbol{w}^*$, and $\xi(\hat{\boldsymbol{w}}, \boldsymbol{w}^*)$ be a performance measure for the learning process. Previous works focused on the *alignment* of weights [2, 4], $\texttt{Alignment} = \hat{\boldsymbol{w}} \cdot \boldsymbol{w}^* / \|\hat{\boldsymbol{w}}\| \cdot \|\boldsymbol{w}^*\|$, measuring the cosine similarity of vectors $\hat{\boldsymbol{w}}$ and $\boldsymbol{w}^*$, i.e., how well the parameters of Alice's reward function are learned. Alternatively, Wilde et al. [5] proposed the relative error in *cost*. We adapt this as the $\texttt{Relative\_Reward} = \phi(\rho(\hat{\boldsymbol{w}})) \cdot \boldsymbol{w}^* / \phi(\rho(\boldsymbol{w}^*)) \cdot \boldsymbol{w}^*$, measuring how much Alice likes the trajectory optimized for $\hat{\boldsymbol{w}}$ compared to the one optimized for $\boldsymbol{w}^*$.

**Problem Statement.** Let $\pi$ be an adaptive policy for designing queries $(P, Q)$, and let $D_K(\pi \mid \boldsymbol{w}^*)$ be the expected set of user feedback when a user $\boldsymbol{w}^*$ is queried by $\pi$ for $K$ iterations. Given a robot motion planner $\rho$, a user with preferences $\boldsymbol{w}^*$, and a budget of $K$ rounds to query the user about their *scale feedback* on two presented paths, our goal is to find an adaptive policy $\pi$ that solves

$$\max_\pi \xi \left( \mathbb{E}\left[ \boldsymbol{w} \mid D_K(\pi \mid \boldsymbol{w}^*) \right], \boldsymbol{w}^* \right). \tag{3}$$

# 4 Approach

We now briefly review learning from choice, and then extend the framework to scale feedback.

## 4.1 Choice Feedback

When presented with two paths $P$ and $Q$, a user returns an ordering $P \succeq Q$ ($P$ is preferred) or $P \preceq Q$ ($Q$ is preferred). In a noiseless setting, we have

$$r(P, \boldsymbol{w}^*) - r(Q, \boldsymbol{w}^*) \geq 0 \iff P \succeq Q. \tag{4}$$

That is, the path $P$ has a reward that is at least as high as that of $Q$ with respect to the hidden true user weights $\boldsymbol{w}^*$. Using $r(P, \boldsymbol{w}) = \boldsymbol{\phi}^P \cdot \boldsymbol{w}$, we can tighten our notation and write $(\boldsymbol{\phi}^P - \boldsymbol{\phi}^Q) \cdot \boldsymbol{w}^*$ instead of $r(P, \boldsymbol{w}^*) - r(Q, \boldsymbol{w}^*)$. Equation (4) already contains an observation model: If the user chooses path $P$, the robot can infer that $P$ has a higher reward with respect to $\boldsymbol{w}^*$. This inequality defines a halfspace $\Lambda(P, Q) = \{\boldsymbol{w} \mid$

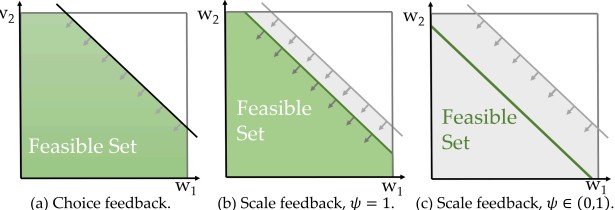

(a) Choice feedback.    (b) Scale feedback, $\psi = 1$.    (c) Scale feedback, $\psi \in (0,1)$.

Figure 2: Different feasible sets learned from choice and scale feedback. Shown is the updated weightspace (green) after observing user feedback for one $(P, Q)$ pair. If $\psi = 1$ scale feedback learns a tighter halfspace; when $\psi \in (0, 1)$ scale feedback learns an equality, i.e., a hyperplane.

$(\boldsymbol{\phi}^P - \boldsymbol{\phi}^Q) \cdot \boldsymbol{w}^* \geq 0\}$ containing all weights that are *feasible* given the observed user choice. Over $k$ iterations, we can intersect the sets $\Lambda(P_1, Q_1), \ldots, \Lambda(P_k, Q_k)$ to obtain the *feasible set* $\mathcal{F}_k$ shown in Fig. 2(a). By definition, this feasible set is convex.

## 4.2 Scale Feedback

Scale feedback allows the robot to gain more information: the robot can also infer by *how much* the user prefers $P$, allowing for learning tighter feasible sets. We extend the model in (4) and show how a noiseless user would provide scale feedback and then study how a robot can learn from it.

**Definition 2** (Maximum Reward Gap). Given a user $\boldsymbol{w}^*$, the maximum reward gap is

$$\delta^* = \max_{P, Q \in \mathcal{P}} r(P, \boldsymbol{w}^*) - r(Q, \boldsymbol{w}^*) = \max_{P, Q \in \mathcal{P}} (\boldsymbol{\phi}^P - \boldsymbol{\phi}^Q) \cdot \boldsymbol{w}^*. \tag{5}$$

We notice that the maximum reward gap cannot be computed, since $\boldsymbol{w}^*$ is unknown to the robot. Nevertheless, we can formulate the user choice model and then derive an observation model.

**User model.** The maximum reward gap helps to define when a noiseless user would indicate a strong preference. We assume this occurs if and only if the difference in reward of $P$ and $Q$ with respect to $\boldsymbol{w}^*$ is at least $\alpha^* \delta^*$ for some $0 < \alpha^* \leq 1$. Here $\alpha^*$ is a saturation parameter which governs

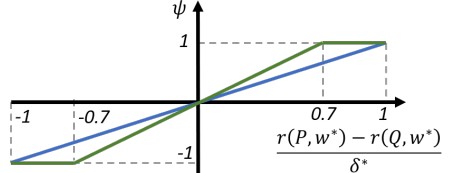
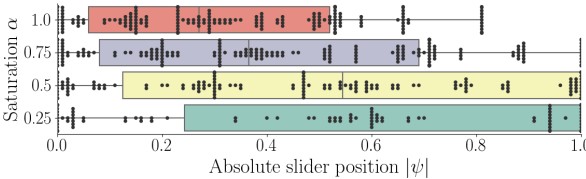

(a) User model for providing scale feedback with $\alpha^* = 1$ (blue) and $\alpha^* = 0.7$ (green).

(b) Example slider feedback for different $\alpha$. The boxplots indicate the four quartiles of the absolute slider values.

Figure 3: Noiseless user model.

at what reward difference (w.r.t. to the maximum gap) the user's feedback gets saturated to a strong preference. For any other $(P, Q)$ where $|(\boldsymbol{\phi}^P - \boldsymbol{\phi}^Q) \cdot \boldsymbol{w}^*| \in [0, \alpha^* \delta^*)$ we assume the user to linearly scale $\psi$ between $-1$ and $1$, which leads to the following model.

**Definition 3** (Noiseless User Model). Presented with two paths $P$ and $Q$, a noiseless user with parameter $\alpha^* \in (0, 1]$ will always provide the following feedback:

$$\psi = \begin{cases} 1 & \text{if } r(P, \boldsymbol{w}^*) - r(Q, \boldsymbol{w}^*) \geq \alpha^* \delta^*, \\ -1 & \text{if } r(Q, \boldsymbol{w}^*) - r(P, \boldsymbol{w}^*) \geq \alpha^* \delta^*, \\ (r(P, \boldsymbol{w}^*) - r(Q, \boldsymbol{w}^*))/\alpha^* \delta^* & \text{otherwise .} \end{cases} \tag{6}$$

We illustrate the noiseless user model in Fig. 3a under different saturation parameters $\alpha^*$. In Fig. 3b, we show a simulated example: for a fixed $\boldsymbol{w}^*$ we simulate how users with different values for $\alpha^*$ would provide scale feedback to the same 20 queries. For larger $\alpha^*$, they position the slider closer to the neutral position. Finally, we derive an observation model for the noiseless user:

$$\begin{aligned} \psi = -1 &\implies r(P, \boldsymbol{w}^*) - r(Q, \boldsymbol{w}^*) \leq \psi \alpha^* \delta^* \\ \psi \in (-1, 1) &\implies r(P, \boldsymbol{w}^*) - r(Q, \boldsymbol{w}^*) = \psi \alpha^* \delta^*, \\ \psi = 1 &\implies r(P, \boldsymbol{w}^*) - r(Q, \boldsymbol{w}^*) \geq \psi \alpha^* \delta^*. \end{aligned} \tag{7}$$

Figures 2(b) and 2(c) illustrate the resulting feasible sets from (7). Moreover, we notice the user-specific and unknown parameters $\alpha^*$ and $\delta^*$ always appear as a product. Thus, we can introduce an auxiliary parameter $\beta = \alpha^* \delta^*$ to write (7) as $\left[ -\psi, \boldsymbol{\phi}^P - \boldsymbol{\phi}^Q \right] \cdot [\beta, \boldsymbol{w}^*] \lesseqgtr 0$. As the model remains linear, the notion of halfspaces and the feasible set $\mathcal{F}$ can be extended to the augmented vector space.

### 4.3 Probabilistic User Feedback

In practice, users are often noisy; they might consider additional or slightly different features than the robot, not follow the linear reward function, or simply be uncertain in some answers. Since we cannot expect users to always provide slider feedback following (6), we introduce a probabilistic model where we add uncertainty to the placement of the slider.

Another practical limitation is the fact that we cannot collect truly continuous feedback from the users. Instead, the slider bar has a step size $\epsilon \in (0, 1]$ such that the user provides feedback of the form $n\epsilon$ for $n \in \mathbb{Z}$ and $-\epsilon^{-1} \leq n \leq \epsilon^{-1}$. Note that $\epsilon \to 0$ retains the continuous scale feedback, whereas $\epsilon = 1$ gives the soft choice model where the feedback is always in $\{-1, 0, 1\}$.

**Definition 4** (Probabilistic User Model). Given a user $\boldsymbol{w}^*$ and a query $(P, Q)$, let $\psi$ be the user feedback defined in the noiseless user model in (6). A probabilistic user using a slider bar with a step size of $\epsilon$ then provides feedback

$$\mu = \text{round}(\psi + \nu, \epsilon) \tag{8}$$

where $\nu$ is a zero-mean Gaussian noise, i.e., $\nu \sim \mathcal{N}(0, \sigma^2)$ with standard deviation $\sigma$, and $\text{round}(x, \epsilon)$ outputs $n\epsilon$ closest to $x$ such that $n \in \mathbb{Z} \cap [-\epsilon^{-1}, \epsilon^{-1}]$.

**Probabilistic Observation Model.** Given the probabilistic user model, we now show how a robot can infer about $\boldsymbol{w}^*$ from scale feedback. In the noiseless case, user feedback defines a feasible set. For the probabilistic case, we instead derive a distribution over $\boldsymbol{w}$ and $\alpha$. Let $\delta(\boldsymbol{w}) = \max_{P, Q \in \mathcal{P}} (\boldsymbol{\phi}^P - \boldsymbol{\phi}^Q) \cdot \boldsymbol{w}$, similar to (5). Then for $0 < \alpha \leq 1$, the belief is defined

$$f(\boldsymbol{w}, \alpha \mid \psi, P, Q) = \begin{cases} \tilde{f}(\boldsymbol{w}, \alpha \mid \psi, P, Q) & \text{if } \psi \in (-1, 1), \\ f^+(\boldsymbol{w}, \alpha \mid \psi, P, Q) & \text{if } \psi = 1, \\ f^-(\boldsymbol{w}, \alpha \mid \psi, P, Q) & \text{if } \psi = -1, \end{cases} \tag{9}$$

where

$$\tilde{f}(\boldsymbol{w}, \alpha \mid \psi, P, Q) \propto \begin{cases} 1 \text{ if } \left[-\psi, \boldsymbol{\phi}^P - \boldsymbol{\phi}^Q\right] \cdot \left[\alpha\delta(\boldsymbol{w}), \boldsymbol{w}\right] = 0, \\ 0 \text{ otherwise .} \end{cases}$$

$$f^+(\boldsymbol{w}, \alpha \mid \psi, P, Q) \propto \begin{cases} 1 \text{ if } \left[-\psi, \boldsymbol{\phi}^P - \boldsymbol{\phi}^Q\right] \cdot \left[\alpha\delta(\boldsymbol{w}), \boldsymbol{w}\right] \geq 0, \\ 0 \text{ otherwise .} \end{cases} \tag{10}$$

$$f^-(\boldsymbol{w}, \alpha \mid \psi, P, Q) \propto \begin{cases} 1 \text{ if } \left[-\psi, \boldsymbol{\phi}^P - \boldsymbol{\phi}^Q\right] \cdot \left[\alpha\delta(\boldsymbol{w}), \boldsymbol{w}\right] \leq 0, \\ 0 \text{ otherwise .} \end{cases}$$

Given noisy user feedback $\mu$ as in (8), we can define a probabilistic density function $f(\psi \mid \mu)$. Together with (9) we derive a compound probability distribution

$$f(\boldsymbol{w}, \alpha \mid \mu, P, Q) = \int_{-1}^{1} f(\boldsymbol{w}, \alpha \mid \psi, P, Q) f(\psi \mid \mu) d\psi. \tag{11}$$

where we can write $f(\psi \mid \mu)$ for $\psi \in [-1, 1]$ as

$$f(\psi \mid \mu) \propto \begin{cases} \Phi\left(\frac{\mu - \psi + \epsilon/2}{\sigma}\right) & \text{if } \mu = -1, \\ \Phi\left(\frac{\psi - \mu + \epsilon/2}{\sigma}\right) - \Phi\left(\frac{\psi - \mu - \epsilon/2}{\sigma}\right) & \text{if } \mu \in (-1, 1), \\ \Phi\left(\frac{\psi - \mu + \epsilon/2}{\sigma}\right) & \text{if } \mu = 1, \end{cases} \tag{12}$$

and $f(\psi \mid \mu) = 0$ for $\psi \notin [-1, 1]$. Here, $\Phi$ denotes the cdf of a standard normal distribution. Finally, given a sequence $D_K = \{(P_k, Q_k, \mu_k)\}_{k=1}^K$ and some prior $f(\boldsymbol{w}, \alpha)$, the joint posterior is

$$f(\boldsymbol{w}, \alpha \mid D_K) \propto f(\boldsymbol{w}, \alpha) \prod_{k=1}^{K} f(\boldsymbol{w}, \alpha \mid \mu_k, P_k, Q_k). \tag{13}$$

Here, we can factor $f(\boldsymbol{w}, \alpha)$ as $\mathbb{P}(\boldsymbol{w})\mathbb{P}(\alpha)$ by assuming $\boldsymbol{w}$ and $\alpha$ are independent and we also have a prior for $\alpha^*$. We then take the expectation of the posterior $f(\boldsymbol{w}, \alpha \mid D_K)$ as our learned user model.

# 5 Algorithm Design

We now outline the learning algorithm. Over $K$ iterations: (i) the robot *actively* generates a query $(P_k, Q_k)$ given previous observations $D_{k-1}$, (ii) the user provides feedback to the query in the form of the slider value $\mu_k$ (in the noiseless case, $\mu_k = \psi_k$), and (iii) the robot updates its dataset $D_k$ using (13). After iteration $K$, the algorithm returns the expected weight $\hat{\boldsymbol{w}} = \mathbb{E}[\boldsymbol{w} \mid D_K]$.

## 5.1 Worst Case Error Bound

To compare scale feedback to choice feedback, we establish a worst case bound on the performance measures for both frameworks. We introduce the *worst-case error* as the maximum negative performance measure, $1 - \xi(\boldsymbol{w}, \boldsymbol{w}^*)$. The constant in front ensures a positive value, which we then discount with the posterior belief, given observations $D$:

$$\texttt{Err}^{\max}(\boldsymbol{w}^*, D) = \max_{\boldsymbol{w}} f(\boldsymbol{w} \mid D)(1 - \xi(\boldsymbol{w}, \boldsymbol{w}^*)). \tag{14}$$

This describes the worst $\boldsymbol{w}$ the robot could pick, discounted by the posterior distribution $f$ learned from data $D$. In the noiseless setting, this simplifies to $\max_{\boldsymbol{w} \in \mathcal{F}} 1 - \xi(\boldsymbol{w}, \boldsymbol{w}^*)$.

**Proposition 1** (Upper error bound). Let $D^S$ denote the observation made from scale feedback and $D^C$ be the observation from choice feedback for the same set of queries. For any user weights $\boldsymbol{w}^*$, it holds in the noiseless setting that $\texttt{Err}^{\max}(\boldsymbol{w}^*, D^S) \leq \texttt{Err}^{\max}(\boldsymbol{w}^*, D^C)$.

The proof follows from the observation $\mathcal{F}^{\texttt{Scale}} \subseteq \mathcal{F}^{\texttt{Choice}}$, i.e., scale feedback removes more volume from the weight set. Hence, the worst choice of an estimate $\hat{\boldsymbol{w}}$ given observations is guaranteed to have a smaller worst case error when using scale feedback. The full proof is in Appendix B.

## 5.2 Active Query Generation

To learn $\boldsymbol{w}^*$ efficiently, the robot chooses the query $(P, Q)$ it presents to the user. While randomly selected queries often lead to some learning progress, actively designing a query can drastically improve learning when the number of iterations is limited. Two recent approaches for learning from choice are information gain [4] and max regret [5]. Information gain seeks to reduce the robot's uncertainty over $\boldsymbol{w}$ while choosing queries that are easy to answer for the user. Max regret, on the other hand, minimizes the maximum regret by showing mutual worst case paths, which also results in easy queries. We leverage both of these methods for our active query generation in scale feedback.

We start with the information gain. Let $H$ denote Shannon's information entropy [36]. As the outcome of the query is yet unknown, a greedy step takes the expectation over $\mu$:

$$\max_{P,Q} H(\boldsymbol{w}, \alpha \mid P, Q) - \mathbb{E}_{\mu \mid P, Q}\big[H(\boldsymbol{w}, \alpha \mid \mu, P, Q)\big]. \qquad (15)$$

We approximate the computation of entropy by summing over a set $\Omega$ of $M$ samples of $(\boldsymbol{w}, \alpha) \sim f$. Thus, following the derivation in Biyik et al. [4], the new query $(P, Q)$ solves

$$\max_{P,Q} \sum_{\mu} \sum_{(\boldsymbol{w},\alpha) \in \Omega} \frac{\mathbb{P}(\mu \mid P, Q, \boldsymbol{w}, \alpha)}{M} \log_2 \left( \frac{M \cdot \mathbb{P}(\mu \mid P, Q, \boldsymbol{w}, \alpha)}{\sum_{(\boldsymbol{w}',\alpha') \in \Omega} \mathbb{P}(\mu \mid P, Q, \boldsymbol{w}', \alpha')} \right). \qquad (16)$$

The max regret policy generates queries $(P, Q)$ such that if the robot learned $P$ but the user optimal solution would be $Q$ is a worst case. With a symmetric perspective over $P$ and $Q$, we have

$$\max_{\boldsymbol{w}^P, \alpha^P, \boldsymbol{w}^Q, \alpha^Q} \mathbb{P}(\boldsymbol{w}^P, \alpha^P \mid D_k) \mathbb{P}(\boldsymbol{w}^Q, \alpha^Q \mid D_k)\Big(R(\boldsymbol{w}^P, \boldsymbol{w}^Q) + R(\boldsymbol{w}^Q, \boldsymbol{w}^P)\Big), \qquad (17)$$

where $R(\cdot, \cdot)$ is the reward difference defined in (2). By observing feedback to such queries it greedily improves the probabilistic worst case error. In contrast to the information gain approach, maximum regret requires $P$ and $Q$ to be optimal trajectories for some users $(\boldsymbol{w}^P, \alpha^P)$ and $(\boldsymbol{w}^Q, \alpha^Q)$. On the other hand, maximum regret does not require a one-step look-ahead and thus no summation over potential feedback values $\mu$, making it computationally lighter.

Equations (16) and (17) now give us two different policies for solving the initial problem (3). In the simulations, we compare how the performance of both benefits from scale feedback.

## 6 Simulation Results

We now present our main simulation results. Additional results can be found in the Appendix.

**Experiment Setup.** We simulate the presented framework using the Driver experiment used in [2, 4–6]. We modify the setup by adding 6 new features, obtaining a more challenging 10-dimensional problem (details on the features, as well as results for the original driver can be found in the Appendix). 71 distinct user preferences $\boldsymbol{w}^*$ are drawn uniformly at random, and each user is simulated with $\alpha^* \in \{.25, .5, .75, 1\}$, making it 284 runs for each method. We set $\sigma = 0.1$ for the noise level. We generate a set of 200 distinct sample trajectories by drawing random weights $\boldsymbol{w}$ and then computing their optimal trajectories. The active query generation methods then optimize over this set. We evaluate learning using the alignment metric and the relative reward.

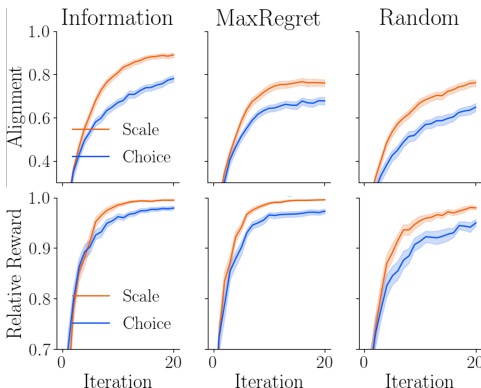

Figure 4: Comparison of scale feedback and soft choice for different active query methods.

As a baseline we use soft choice (strict choice showed a slightly poorer performance). To ensure a fair comparison, we emulate soft choice by setting the step size to $\epsilon = 1$ and use the same noise model for both forms of feedback.

**Results.** Fig. 4 shows the alignment and relative reward for the driver experiment for information gain, max regret and random query generation. We observe that in all cases scale feedback significantly improves the performance over soft choice in both metrics ($p < .001$ in all cases with two-sample $t$-test). When using the proposed scale feedback, the alignment after 20 iterations improves from .77 to .86 for information gain, from .67 to .76 for max regret, and from .64 to .75 for random queries. The relative reward improves for information gain and max regret similarly from .97 to 1, i.e., the learned solution is optimal. Both methods make most progress during the first 10 iterations. Random queries improve the final relative reward from .94 to .97. Overall, the simulation showcases that scale feedback improves learning, independent of the query selection method. For information gain and max regret, scale feedback allows for finding optimal solution, i.e., collecting 100% reward, within a small budget of iterations. In Appendix D, we show additional simulation results for higher noise.

# 7 User Study

Finally, we analyze the scale feedback in comparison with choice feedback and under different active querying methods with two user studies.[1] In both studies, we used $\epsilon = 0.1$ for scale queries.

**Experiment Setup.** We designed a serving task with a Fetch robot [37] as shown in Fig. 1, and generated a dataset of 120 distinct trajectories. Human subjects were told they should train the robot to bring the drink to the customer in the manner they prefer, paying attention to the following five factors: the drink (out of 3 options) to be served, the orientation of the pan in front of the robot, moving the drink behind or over the pan, the maximum height of the path, and the speed. The subjects were also informed about the types of queries they will respond to.

**Independent Variables.** In the first experiment, we wanted to compare scale and soft choice under random querying, and scale under random and information gain querying. Hence, we varied the query type and the querying algorithm among: (i) soft choice with random querying, (ii) scale with random querying, and (iii) scale with information gain querying. In the second experiment, we wanted to compare scale and soft choice under information gain querying. Hence, we employed: (i) soft choice with information gain querying, and (ii) scale with information gain querying. For all, we took $\sigma = 0.35$ based on pilot trials with different users (see Appendix E).

**Procedure.** We recruited 18 participants (5 female, 13 male, ages $20 - 55$) for the first, and 14 participants (5 female, 9 male, ages $20 - 56$) for the second experiment. Due to the pandemic conditions, the subjects participated in the study remotely with an online interface as in Fig. 1. The study started with an instructions page with a two-question quiz to make sure the participants understood how to use the interface. After reading the instructions, we had the subjects fill a form where they indicated their preferences for each of the five individual factors described above, to encourage them to be consistent in their responses during the data collection.

In the experiments, each participant responded to 10 queries generated with each of the algorithms. After each of these 10-query sets, they were shown the optimal trajectory from the dataset with respect to their learned reward function. The participants responded to a 5-point Likert scale survey (1-Strongly Disagree, 5-Strongly Agree) for this trajectory: "The displayed trajectory fits my preferences on the task." We also collected scale feedback for 10 more randomly-generated queries for validation in each experiment. We randomized the order of these sets (of 10 queries) to prevent any bias. The interface provided a "Sync Videos" button to restart both videos for easier comparison.

**Dependent Measures.** As an objective measure of the learning performance, we calculated the log-likelihood of the validation set (of 10 scale queries[2]) under the posterior $f(\boldsymbol{w}, \alpha \mid D)$ learned using the 10 queries generated via each algorithm, i.e., we calculated:

$$\texttt{Log-Likelihood} = \log \mathbb{P}(D_{\text{validation}} \mid D) = \log \mathbb{E}_{\boldsymbol{w}|D} \left[ \mathbb{P}(D_{\text{validation}} \mid \boldsymbol{w}) \right] \tag{18}$$

We also used the responses to the 5-point Likert scale survey questions to measure how well the learned rewards achieve the task. Finally, the users took a post-experiment survey where they rated (from 1 to 5) the easiness and expressiveness of soft choice and scale questions.

**Hypotheses.** We test the following hypotheses.
**H1.** *Scale feedback leads to faster learning than soft choice feedback.*
**H2.** *Querying based on information gain accelerates learning compared to random querying.*
**H3.** *Users will prefer information gain over random querying in terms of the optimized trajectories.*
**H4.** *Users will prefer scale feedback over soft choice feedback in terms of the optimized trajectories.*
**H5.** *Users will rate the scale feedback as easy as soft choice feedback.*
**H6.** *Users will rate the scale feedback as expressive as soft choice feedback.*

**Results.** We present results of the first and the second experiments in Figs. 5 and 6, respectively. It can be seen that the log-likelihood of the validation set after learning the reward function via scale feedback is higher than learning via soft choice feedback, under both random and information querying. Besides, information gain based query generation accelerates the learning and leads to higher log-likelihood values compared to random querying. All of these comparisons are statistically significant with $p < .001$ (paired-sample $t$-test), so they strongly support **H1** and **H2**.

---

[1] We have IRB approval from a research compliance office under the protocol number IRB-52441. A summary video is at https://sites.google.com/view/reward-learning-scale-feedback, and the code at https://github.com/Stanford-ILIAD/reward-learning-scale-feedback.

[2] We present results with a validation set that consists of both scale and soft choice feedback in Appendix F.

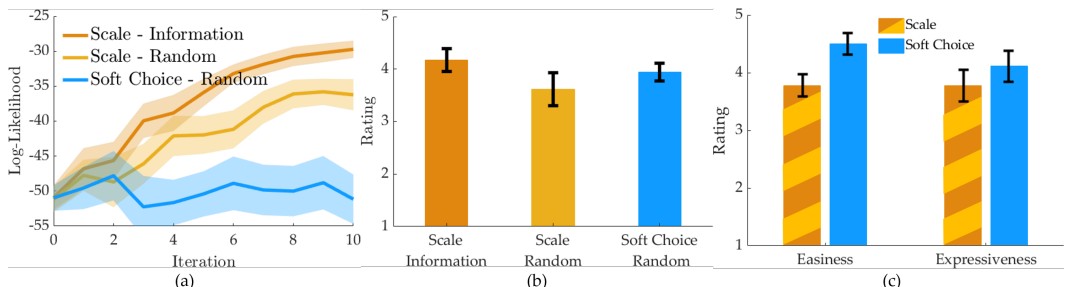

Figure 5: All results are shown for the first experiment (mean±s.e. over 18 subjects).

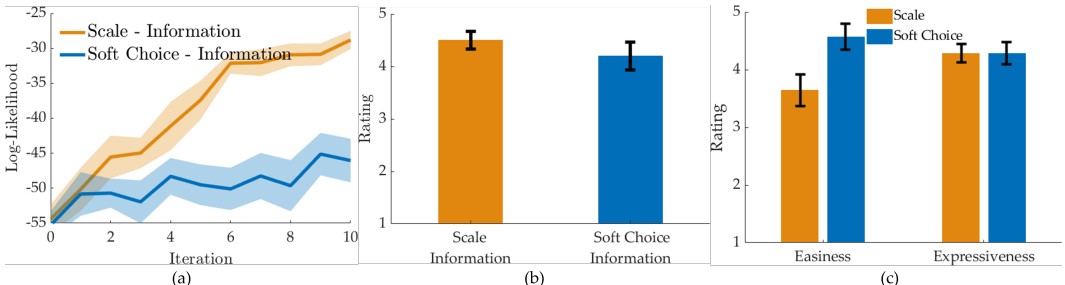

Figure 6: All results are shown for the second experiment (mean±s.e. over 14 subjects).

In Fig. 5(b), it can be seen active querying led to learning reward functions that better optimize trajectories compared to random querying – this comparison was somewhat significant with $p \approx .05$, supporting **H3**. In fact, when we fit a Gaussian distribution to the ratings, we observe that it is 1.95 times as likely to get a better rating with information gain querying than random querying. Surprisingly, learning via soft choice achieved slightly higher reward than learning via scale when queries were randomly selected, and slightly lower reward when queries were generated based on information gain. However, these comparisons are not statistically significant. This is indeed analogous to the relative reward comparisons in Fig. 4: more complex tasks might be needed to better analyze the difference between the two methods. Thus, we neither reject nor accept **H4**.

Finally, the subjective results in Fig. 5(c) and 6(c) suggest that users find the soft choice feedback slightly, but consistently, easier than the scale feedback ($p < .01$), rejecting **H5**. This is not surprising, as it is often easier to make a pairwise comparison and the "About Equal" option in the soft choice questions makes them even easier [4]. On the other hand, there was no statistically significant difference in terms of expressiveness of scale and soft choice feedback, partially supporting **H6**. In summary, it is interesting that our users perceived the soft choice as easier and even more expressive at times; even though quantitatively, the scale feedback significantly outperforms the soft choice.

## 8 Discussion

**Summary.** We proposed scale feedback for reward learning where users provide more nuanced feedback than choice. We introduced a user model and showed how a robot can infer reward from noisy scale feedback. We adapted state-of-the-art query generation methods to accelerate learning. In simulations and a user study, scale feedback significantly improved learning. Users rank choice feedback as slightly easier, but both forms of feedback as equally expressive. However, the minor decrease in ease of use is out-weighted by a strong improvement in learning performance.

**Future Work.** We proposed scale queries as a way to give nuanced feedback between two trajectories. It is possible to extend them to $n + 1$ trajectories, with specialized user interfaces that allow users to select a point from an $n$-simplex instead of a slider bar. Future work should investigate this and if users can still give reliable feedback to these more complex queries.

In our experiments, we used a pre-computed trajectory set. Alternatives, e.g., optimizing queries over action sets as in [2], or using planners as in [5], should be studied for real-time online learning systems. The high estimate of $\sigma$ in the user studies suggests the proposed probabilistic model may be inaccurate. Future work should refine the user model, including interactively learning $\sigma$; or fit a new user model that does not necessarily adopt a Gaussian noise. Surprisingly, users did not perceive scale feedback as more expressive. This could be addressed with improving interface design as well as designing a query generation method that actively exploits the slider's expressiveness.

**Acknowledgments**

This research is partially supported by the Natural Sciences and Engineering Research Council of Canada (NSERC). The authors would also like to acknowledge funding by NSF grants #1849952 and #1941722, FLI grant RFP2-000, and DARPA.

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
