# OpenReview forum: "Learning Reward Functions from Scale Feedback"
_robot-learning.org/CoRL/2021/Conference — CoRL2021 Poster_

### Official Review · Reviewer_2eoP · 2021-07-22

**Originality:** Good
**Technical Quality:** Good
**Clarity Of Presentation:** Excellent
**Impact:** 2

**Recommendation:**

Weak Accept: I recommend accepting the paper, but will not argue for my recommendation if the majority of other reviewers have a different opinion.

**Summary:**

I have updated my estimate of the technical quality from fair to good in response to further user studies and more detailed analysis of the data. Overall, I have upgraded my judgment from Strong reject to weak accept in response to this further evidence. I urge the authors to perform further experiments in order to confirm or reject their understanding of the discrepancy between log-likelihood and user ratings, and I personally still find it more likely that there is some problem in the log-likelihood evaluation. However, the collection of evidence is interesting enough that it ought to be published. The original review is below.

===

Choice feedback has been successful as an alternative way to specify the desired behavior of an RL agent. The key feature of choice feedback is that it is easier to correctly produce compared to optimal demonstrations or a manually specified reward function. However, choice feedback is easy to produce in part because it is an inherently low information communication mechanism: each binary choice by definition provides at most one bit of information.

Humans can sometimes provide more information than this single bit when comparing two trajectories. For example, in prior literature there are methods which present the human with a third option to indicate that two trajectories are roughly as good as each other. This paper takes this insight to it's natural conclusion by allowing the user to specify a scalar strength indicating precisely how much they prefer one trajectory to another, including not at all. Whereas most (perfect, noiseless) choice feedback implicitly imposes a halfspace constraint on the reward function space (under known linear reward features), the paper shows that a scalar opinion reduces the set of possible reward values to a hyperplane in the reward space (reducing the total dimension of the feasible rewards by one). Humans are never actually capable of providing noiseless feedback, so the paper also develops a natural Gaussian noise model of the probability of a human reporting a certain preference strength under a given ground truth reward function, which they use to infer the true reward function in a Bayesian fashion.

Human feedback elicitation research often operates by picking a feedback modality to elicit from the human, a Bayesian model of how that feedback is produced, and then finally a criterion with which to determine the optimal query to ask at every timestep. This paper is not attempting to innovate on the latter, and so evaluates their method using simulated and real human data under two such criteria, maximizing expected information gain and minimizing the maximum regret, as well as a baseline where completely random questions are asked independent of the current belief distribution. Experiments are performed using all of these query generation methods under a binary choice feedback modality and the new scalar feedback modality in the simulated case, and a smaller subset of these experiments are performed on human data.

**Issues:**

How exactly was the choice of $\sigma=0.35$ made in the user study (as mentioned on line 255)? Shove this in the appendix if you have to, but this is an important part of your experimental method. Did you try using a per-user sigma value? In my experience different users will behave wildly differently in these kinds of settings. But perhaps there simply wasn't enough data per user to determine a meaningful $\sigma$ value.

On line 287, tell me exactly what the p-value is. Given how small all of the other p values are, I'm not inclined to trust a "tending towards significance" conclusion. You can say that the study is underpowered and H4 is inconclusive as you do with H3 (for which I would also like a p-value for the hypothesis that you're exactly wrong). Alternatively, if you think the current results really do suggest there's something there that shouldn't be ignored, you can do a Bayesian analysis and report a (presumably) moderate but not overwhelming likelihood ratio, which would say what I imagine you're trying to say with "$p \approx 0.05$" but in a principled way.

The choice of experiments in the user study is very confusing, and results in a weak baseline that makes it impossible to be certain that scale queries are robustly better than soft choice queries. In the prior work on the subject (I'm mostly familiar with Biyik and Sadigh's work) the rough ranking of the efficacy of active query generation criteria is infogain > regret > random. In some sense random queries are the obvious thing to do, and serve as a baseline for all other active methods. Why then is the only soft choice query baseline in the user study using questions generated randomly, instead of one of these better criteria? Due to this choice, the only apples-to-apples comparison we can make is between Scale - Random and Soft Choice - Random, leaving completely open the possibility that soft choice might outperform scale when both are using infogain generate queries. Even if this is not the case (i.e. scale infogain queries are the best), it is interesting to know how relatively important the active reward criteria is relative to the form of query. If soft choice infogain outperforms scale random queries, then we know it is most important to use infogain queries (or possibly some other non-random criteria), and that scale queries are a robust improvement, but perhaps a minor one. It is hard to tell from the figures in the paper (I would appreciate a table of the exact final alignment, relative reward, and log likelihood values of all runs in the appendix), but this is plausibly true in the simulated experiments as well. Alternatively, we mind find that scale random outperforms soft choice infogain, and that the choice of query type is more important. As it stands, there are many fundamental questions about your method that we cannot answer due to the choice of a weak baseline.

This may just be my familiarity with the area, but Proposition 1, while informative, also seems essentially trivial. It would significantly improve the paper if there were a similar theoretical result for the noisy case, even a noisy case where you have to assume "too much" about the noise model in order to get a result.

Several plots in figures 10 and 11-13 make it appear as if choice queries are approaching the performance of scale queries. If this is the case scale queries may be much better for requiring less data, but it would be interesting to see where this saturation point is. Based on the runs where the two lines do meet, and due to your theory, it seems unlikely that choice ever surpasses scale, but this would be valuable information to have in future comparisons of the information value and human difficulty of different styles of query.

There are a couple of places in the introduction where I agree with the spirit of the claim but disagree in the specifics:

Line 51: "While demonstrations are usually most informative" Demonstrations are often not more informative than preferences. In particular, optimal demonstrations are often good along all features, and so it can be in principle impossible to disambiguate between different relative weights between pairs of features which are highly correlated in optimal trajectories. To steal the example from your video, if we only see examples of passing the beverage that are fast and don't break anything on the table, it is in principle impossible to tell if the speed or the safety is the more important factor here.

Line 58: "it also only provides one bit of information" choice queries provide AT MOST one bit of new information, they often provide much less, especially if the query being asked is essentially redundant considering prior queries. I personally find this is often the case when working with active preference elicitation algorithms. This only strengthens your point.

The following are grammatical or formatting issues that I expect can be fixed easily and are primarily for the author's use:

Line 162: Please write out all of the dot products explicitly as equalities or inequalities. At the very least, place commas as separators between the terms, the spacing alone makes it hard to tell.

Line 240: "the simulation showcase that" -> "the simulation showcases that"

Figure 5: The lines below and above the figures are distracting

**Reviewer Expertise:**

Excellent: Expert knowledge on the topic of the paper

**Strengths And Weaknesses:**

Strengths:
1. The idea to elicit scalar preferences from a human is a natural extension to existing successful ideas in choice feedback, and frankly I wish I had done these experiments myself.
2. The paper is clearly written with excellent figures.
3. The appendix includes a large number of supporting simulated human experiments.

Weaknesses:
1. The experimental results of the paper do not support the conclusions stated in the abstract and introduction.
2. There is an important missing experiment from the human study.
3. The practical choice of $\sigma$ is an important complication of the method is not well discussed.

**Summary Of Recommendation:**

As the paper stands the experiments can only support the conclusion that the Gaussian noise model under scale feedback is not so incorrect as to be disastrous. There is no robust evidence to suggest that scale feedback is superior to choice feedback on human data. The main issue is that no choice experiment was performed using a state of the art active query criteria (e.g. infogain). This makes it impossible to make like to like comparisons between scale and choice feedback while using the techniques either are likely to be used with. The results reported from the experiments that were performed support neither the claim that humans find scalar feedback more expressive nor the claim that the learned reward function under scalar feedback better fits their preferences on the experimental task. In fact, on the latter measure, the evidence suggests that humans will likely prefer the reward function learned under choice feedback, and they prefer choice feedback when comparing like query generation to like. As the likelihood models are different for the scalar and choice feedback, log likelihood cannot be compared between feedback types, and so I cannot see it as an indication of learning good reward functions. In order to accept this paper I would need to see the missing infogain + choice human experiment and either

a) results indicating that scalar feedback robustly outperforms choice feedback

b) if scalar feedback is not always better, a discussion of when/for what problems it is, or

c) a more frank portrayal of the results, perhaps by saying that you can get to the same (or a slightly worse) reward function in many fewer queries to the human, or whatever conclusion the results of the final experiment support.

---

> ### Author Response · Authors · 2021-08-26
> **Response to Reviewer 2eoP**
>
> We thank the reviewer for their incredibly helpful and insightful comments. We carefully considered all the points raised by the reviewer and conducted a new user study so that our results are complete. We made the changes in the paper with blue ink for the convenience of the reviewer.
>
> **[LOG LIKELIHOOD COMPARISON BETWEEN DIFFERENT QUERY TYPES]**
>
> We first want to clarify why log likelihood can be compared between different query types. In both simulations and user studies, we first learn a posterior (for the reward) using each query type. We then collect a validation set of randomly generated scale queries. Finally, we compute the likelihood of this same validation set under each posterior. As the validation sets are the same, the likelihood values give us a comparable metric. The validation set consisting of scale queries does not introduce a bias, because the posteriors are directly over the reward weights and a good posterior should be able to predict any feedback type. The same approach was used in [3] to compare different querying methods. [8] has also used a similar metric.
>
> **[NEW USER STUDY]**
>
> The reviewer rightfully says showing scale feedback outperforms soft choice feedback under random querying does not necessarily mean it will also outperform under information gain querying. Thus, we conducted a new user study. The details can be found in the updated paper. Briefly, we replicated the old experiment with scale and soft choice queries, both under information gain querying.
>
> The new figure (Fig. 6) shows scale feedback leads to faster learning even with information gain querying (Fig. 6a). The comparison in the final log likelihood is statistically significant ($p<.001$). Besides, Fig. 6b presents a result similar to Fig. 5b: from the users’ perspective, the final performances of the robots that learned using scale and soft choice are comparable. In fact, there was a slight preference for scale feedback in the new experiments, but none of these two comparisons (under different querying) is statistically significant. Finally, Fig. 6c is similar to Fig. 5c: Users rate the soft choice queries as easier, but there is no significant difference in terms of expressiveness.
>
> **[$\sigma$ SELECTION IN THE USER STUDIES]**
>
> We added a new Appendix section, describing how we selected $\sigma=0.35$ for user studies. Briefly, we collected data from 3 users who did not participate in the other user studies by asking them randomly generated scale and soft choice queries. We divided this data into training and validation sets, and found the $\sigma$ that best explained their behavior in the validation set.
>
> As the reviewer suggests, per-user sigma values could work better, but it would require more data from all users. Due to this, we used a single $\sigma$.
>
> **[$p$-VALUES]**
>
> We now report the $p$-values the reviewer asked for. The comparison between “Scale-Information” and “Scale-Random” in Fig. 5(b) for H3 has $p=.0531$. We reported this as $p\approx .05$. In the new version, we also included a Bayesian analysis: “when we fit a Gaussian distribution to the ratings, we observe it is $1.95$ times as likely to get a better rating with information gain than random querying”.
>
> For H4, in Fig. 5(b) where we compare “Scale-Random” and “Soft Choice-Random”, $p=.1429$, which does not indicate a significant difference. In fact, in the new results where we compare “Scale-Information” and “Soft Choice-Information”, the mean rating is higher for the scale feedback. However, $p=.1825$, which is again inconclusive.
>
> **[TABLE OF ALL RUNS]**
>
> As suggested by the reviewer, we added two tables to the Appendix (Tables 2 and 3) to report the numerical values of all plots in the main paper.
>
> **[SATURATION POINT BETWEEN SCALE AND SOFT CHOICE]**
>
> Choice feedback would indeed reach the performance of scale feedback with enough data. Intuitively, a response to a scale query gives more information about the reward difference. However, if we make multiple choice queries (of the same trajectory pair), then we will know with what ratio the user selects the first trajectory (under the probit model). This ratio will also give more information about the reward difference, which explains why the methods can converge to the same posterior. And the saturation point depends on multiple factors, e.g., user’s parameters ($\sigma$ and $\alpha$), the step size of the slider ($\epsilon$), and the features of the task, which directly affects the response probabilities.

---

> > ### Author Response · Authors · 2021-08-26
> > **Response to Reviewer 2eoP (cont.)**
> >
> > **[EXTENDING THE THEORETICAL RESULTS TO NOISY HUMANS]**
> >
> > Theoretical results do not extend to the noisy case, because adversarial users and/or tasks can be constructed. Imagine a task where the user’s noiseless response $\psi\in\\{-1,0,1\\}$. With soft-choice, under moderate noise, the true response can often be recovered. However, when we go to scale feedback with $\epsilon=0.5$, the noisy responses $-0.5$ and $0.5$ introduce additional uncertainty, potentially causing performance loss. While this is an edge case, we believe a task/user-specific result, where we assume some true distribution for $\psi$, would not be as general and would need to be carefully justified.
> >
> > **[OTHERS]**
> >
> > We also thank the reviewer for their discussion about the claims on which they disagree about the specifics. We fixed those sentences along with the typos.

---

> > ### Comment · Reviewer_2eoP · 2021-08-31
> > **A confusing set of results**
> >
> > I apologize for the slow response, I have been struggling with how to interpret the new evidence.
> >
> > I appreciate that the new user study was performed. This plugs a large gap in the evidence base.
> >
> > I did not realize that log-likelihood was being computed using the same validation dataset for each method. I agree this is more directly comparable. In principle a good reward posterior should have high likelihoods for human preferences of any form, but in general the different human biases present in different forms of question may cause the posteriors to "overfit" to these biases, which will also be present in the validation dataset. One could imagine a validation strategy where questions are asked of the human from each method, and the likelihood of their answers under each posterior is aggregated. This isn't a perfect metric, as some forms of question might be inherently noisier/more biased than the others. Intuitively I wouldn't have expected scale vs choice questions to be different enough for this to matter in this case, but the rest of the evidence conflicts with the log-likelihood comparison, and so it's difficult to trust a potentially flawed metric in comparison.
> >
> > My understanding of the evidence is as follows, and I invite you to make corrections to this: After 10 iterations of updating the reward posterior, the user ratings of trajectories that result from optimizing a reward from this posterior are statistically indistinguishable. The posterior distributions generated from this training are better at explaining a holdout set of scale queries, which is evidence that the scale likelihood model is internally consistent, but I cannot be certain that it is evidence that scale queries are better than soft choice queries at generating a reward posterior. Users do not feel that scale queries are more expressive, although they do feel that they are harder to produce, which is consistent with needing to provide more bits to generate an answer. I see two interpretations that are consistent with the evidence in aggregate: One is that scale queries do not produce higher quality reward posteriors than soft-choice queries on the environment tested, and that the log-likelihood evaluation procedure is biased in the way I've described above. The other is that 10 iterations are enough for all methods to saturate, and although the log-likelihood of the scale method's posterior is robustly higher, in absolute terms the difference is not enough to affect the quality of the produced trajectories. I'm leaning towards the former, as the likelihood of soft-choice random appears to go down overall, but still has decent user ratings, implying that the likelihood is not at all capturing whatever learning soft-choice random is doing.
> >
> > I appreciate the additional details regarding $$\sigma$$ selection, p-values, and the additional tables. I agree that saturation likely depends on too many variables to give a meaningful conclusion.

---

> > > ### Author Response · Authors · 2021-08-31
> > > **More Analyses on the Validation Set**
> > >
> > > We appreciate the time the reviewer took to go over our changes and make additional comments. We are happy to hear that the new study plugs a large gap in the evidence base, and we agreed with the reviewer in all the points but the validation set.
> > >
> > > From the reviewer's comments, we understand that the reviewer and we are in complete agreement in terms of why it is ideally okay to employ a validation set of a single query type: a good posterior should be able to predict any form of feedback. However, the reviewer rightfully says that humans have biases: although a unimodal validation set is okay under the user choice model we assumed, humans do not necessarily follow that model. Hence, the reviewer suggests an alternative way of measuring log-likelihood where the validation set consists of both scale and soft choice queries.
> > >
> > > Given that there are only a few hours to the deadline, we cannot perform a new complete user study. However, we were able to make additional analyses using the human data we already collected. We present this new evidence in Appendix G. Below, we summarize our procedure and findings.
> > >
> > > As a reminder, we collected 4 different data sets in our first user study:
> > > - 10 scale queries that are actively generated with information gain
> > > - 10 randomly generated scale queries for training
> > > - 10 randomly generated soft choice queries
> > > - 10 randomly generated scale queries for validation
> > >
> > > In this new analysis, we took the first 7 queries of the first three sets: so each training set has only 7 data points now. We then aggregated the remaining 3 queries of the third set (randomly generated soft choice queries) with 3 queries of the fourth set (the original validation set that consists of randomly generated scale queries). In this way, we obtained a validation set that is completely separate from the training data and consists of equal number of scale and soft choice queries.
> > >
> > > We then measured the log-likelihood under this new validation set for up to 7 training data points. Figure 14 in the Appendix shows the results. Even when the validation set consists of a mixture data, we have the same trend: "Scale - Information" performs the best, and "Scale - Random" still outperforms "Soft Choice - Random". To be more precise, "Scale - Information" and "Scale - Random" both outperform "Soft Choice - Random" with statistical significance ($p < 0.05$). This shows the scale feedback indeed outperforms the soft choice feedback in terms of log-likelihood -- the reason for the results is not the bias in the validation set.
> > >
> > > Unfortunately, we are able to perform this additional analysis only with random querying; because our second user study, where we studied information gain querying, did not collect any randomly generated soft choice feedback (and information gain queries would have their own biases due to the way they are generated).
> > >
> > > Although this analysis shows the superiority of scale feedback over soft choice feedback in terms of log-likelihood, it does not answer the question why user ratings did not have a significant difference between the two feedback. The answer to this question requires more analysis and possibly more data collection. However, we speculate the following reason, which is in the same lines as the reviewer's second interpretation: the mean user ratings are always around $4$, and even higher than $4$ when queries are actively generated with information gain. This means the users are happy with the optimized trajectories, so we can say that $10$ queries are enough in this task to find the optimal trajectory. However, while user ratings measure how close the optimal trajectory with respect to the robot's posterior is to the optimal trajectory the user has in mind; log-likelihood measures the predictive performance of the posterior. Therefore, having a high user rating does not necessarily mean the robot can accurately compare two suboptimal trajectories. On the other hand, a high log-likelihood value indicates good predictive performance, which is crucial in many robotics applications, such as behavior modeling. Hence, we claim: (i) learning from scale feedback improves the predictive performance over learning from soft choice feedback, and (ii) a more complex task might be needed to show scale feedback leads to more efficient learning than soft choice feedback, which is also suggested by our simulation studies.

---

### Official Review · Reviewer_pmpv · 2021-07-22

**Originality:** Good
**Technical Quality:** Good
**Clarity Of Presentation:** Good
**Impact:** 3

**Recommendation:**

Weak Accept: I recommend accepting the paper, but will not argue for my recommendation if the majority of other reviewers have a different opinion.

**Summary:**

This paper introduces the use of scale feedback to learn user reward functions expressed as a probabilistic model. The author empirically evaluate the advantages of their method in simulation and in a real robot experiment.

**Issues:**

I would appreciate if the authors can help me understanding the impact and originality of their work. For example, why do you think this has not been proposed in the past? What are the major problems that other researchers may have encountered?

**Reviewer Expertise:**

Poor: Limited knowledge of the area

**Strengths And Weaknesses:**

Strength
======
- the proposed method is intuitive and sound;
- the lower bound on the worst case error is a nice result;
- experimental evaluation is well described and informative.

Weaknesses
==========
- I find the technical description of the method a bit hard to follow;

**Summary Of Recommendation:**

I like how the paper motivates the approach and provides a solid theoretically sound algorithmic implementation. I am not an expert in this field of study, but I find the validity of this method quite intuitive. In fact, I find it weird that it has not been proposed in the past. I am not knowledgable of the related literature, so I will trust the opinion of other reviewers about the originality of this work. It’s also difficult for me to judge the impact of this work in the robotics community, but I definitely think it could be interesting to a group of researchers. Overall, I am convinced of the validity of this work thanks to the accurate mathematical formalization of the problem and the intuitive motivation of the proposed method, although I am not sure about its impact and originality.

---

> ### Author Response · Authors · 2021-08-26
> **Response to Reviewer pmpv**
>
> We thank the reviewer for their comments. We are glad to hear that the reviewer states that our method is sound, we have nice results and the experimental evaluation is well-described and informative.
>
> We agree with the reviewer that it is interesting that scale feedback was not proposed before. Being the researchers who also recently came up with this idea, we speculate that the reasons might be some of the following:
>
> 1- Most works in  preference-based learning would discard the data when the user was unable to compare two items (trajectories in our case). However, it was recently shown by [4] that users can be allowed to state “About Equal” and such responses can actually be used to better learn the reward function, and so the data-efficiency increases. Perhaps, this was the step required to realize a more nuanced and more informative response model is possible and useful.
>
>  2- Most preference-based reward learning works in the robotics community use the softmax model (also known as the multinomial logit model) to mathematically characterize the users’ response probabilities. This implicitly requires the number of possible responses to be equal to the number of trajectories in each query (unless the system designers make very specific modifications as in [4]), which would make the scale feedback hard to use. To the best of our knowledge, even though the probit response model, the model which our work adopts, has been used in other preference-based learning literature, it has not been as common as the softmax model in robotics (possibly because the softmax model appears in many reinforcement learning algorithms).
>
>  3- As our user studies suggest, scale questions tend to be more difficult than the choice questions. This might also be another reason why researchers did not investigate it. However, we discovered that this decrease in the ease of use is out-weighted by a strong improvement in the learning performance.

---

> > ### Comment · Reviewer_pmpv · 2021-08-31
> > **Response to Authors' response**
> >
> > I thank the authors for their response that helped me clarify some doubts about this work. I still think this work is interesting and I keep my score, although my confidence about it has not increased. I'll not argue for acceptance if the majority of reviews will tend towards rejecting this paper.

---

### Official Review · Reviewer_cvuG · 2021-07-23

**Originality:** Fair
**Technical Quality:** Fair
**Clarity Of Presentation:** Good
**Impact:** 3

**Recommendation:**

Weak Reject: I recommend rejecting the paper, but will not argue for my recommendation if the majority of other reviewers have a different opinion.

**Summary:**

Update: I sincerely appreciate the authors' thorough and thoughtful responses as well as updates to the research/paper itself. I have updated my review accordingly.

======
This paper presents a framework for learning user reward functions for humanoid robotic tasks (and possibly other more general robotic applications) by using scale feedback in lieu of binary or ternary choices. The paper presents the framework in terms of a noiseless user model as well as a noisy user model, and it also provides theoretical results on error bounds for some of the framework. The paper presents an algorithm for query generation, and then evaluates the efficacy of the algorithm relative to several baselines, across two studies.  The paper is clearly written and structured well.



**Issues:**

I am not entirely clear what the first paragraph (lines 14-18) have to do with the paper's focus and contributions. Yes, robots need to trade off between goals and have competing objectives, but I believe the paper is about understanding user preferences. While these concepts are ultimately related in any interactive robotic environment, I do not understand this paragraph's purpose.

Regarding scale feedback as a general notion, I have one general concern and question. Wouldn't it be more benefiical to compare more than 2 trajectories, as opposed to having a sliding scale between a binary choice? This latter approach still seems very simplistic and might not help a human think about all the amazing possibilities that a robot could identify.

Line 87, delete comma after "setting. (I think)

How senstive is the system's performance to the annotations on the slider bar? There are 3 annotations in the approach; what if there were 5? And are they anchored in any type of way, such as what is typically done in polling and the like?

Can the authors provide more justification for the modeling assumptions in definition 4? In particular, why is user noise zero-mean Guassian? And why is it stationary? Why is this stationarity independent of what the actual choice psi is? For example, I imagine the choice 0 might generally be less certain than 1 or -1. Thoughts here?

Proposition 1 is well-reasoned. However, this seems like a bit of a "strawman". Of course if you give more options to non-noisy agent, any algorithm should do better both in the average case and worst case. The authors seem to imply this with the point that adding choices removes volume from the weight set. But what happens if the human is noisy?

Line 204 states that max regret "hand minimizes" the maximum regret. Is this a typo, or a term I am unfamiliar with?

Line 238-239: missing word: "...make most progress *during* the first 10 iterations"?

Figure 5(b) is very interesting. The authors discuss this a bit in the body text, but I am not sure it is helpful. Can the authors at least propose a framework that would help (in future work) to support or invalidate H3? Is it as simple as doing more experiments? Or does the design of experiments actually need to be changed?

I do not agree with the conclusion, in line 297, that the data partially supports H5. It either leads one to reject H5, or partially reject it, but not partially support. Furthermore, what does "expressiveness" even mean? Was this explained tot he user during the questioning?

Regarding the discussion about expressiveness: returing to my earlier comment, I think the important factor to the user is the expressiveness of the *trajectory itself*. That is, do the presented trajectories really represent the tradeoffs I care about? This is why I had the earlier question about presenting multiple "classes" of trajectories. What is a user really expressing when they say they prefer one trajectory over another? There are so many different objectives (as the authors acknowledge), and collapsing things down to a single dimension - which a sliding bar does, no matter how small epsilon becomes - severely limits the user's ability to tell the machine about his/her preferences.

In short, I am skeptical about whether the slider even fundamentally allows the system to "ask the right question" to the users. Going from a binary (or 3-nary) choice to an infinite slider is fine (apparently, from the data) if one wants to accurately estimate a user's optimal policy. Therefore, who really cares about expressiveness? In the end, how much does a user's opinion of the interface matter, as long as the user can stay engaged to the point that a model can be learned?

**Reviewer Expertise:**

Good: General knowledge of the area

**Strengths And Weaknesses:**

The paper focuses on an interesting problem of understanding user preferences and then being able to formally encode this information in terms of learning a reward function. The paper is structured well, the prose is clear.

My main suggestion has to do with scope, and I am not sure if this is "fair" given that I would essentially be asking the authors to change their approach entirely. I question whether a slider bar (scale response) evaluating two different trajectories is even the right way to think about evaluating/understanding/learning/encoding user preferences. I have several questions about this in my detailed comments below. Reducing everything down to a binary choice does not seem like a great idea, as the paper points out well. However, turning this choice into a sliding scale does not seem to enhance thing much, and as the users note, this does not seem to be very expressive. While the results suggest that the method does indeed converge on a more accurate reward function, I think one of the promises (and challenges) of explicitly including humans involves the now-overused terms like "explainability" and "transparency" and "trust" and things of this ilk. If the method is really just about learning rewards, then I suppose this paper does well enough. Again, I recognize that this assessment might not be fair and I am trying to be helpful here, I promise. What is the purpose of bringing the human in the loop? Can the authors make this more clear, given the relatively simple task that was used for evaluation/experimentation? To me, the big topics involving humans and automation have to do with the buzzwords I mention above, not necessarily understanding user preferences for a task that (in my opinion) is amenable to just writing down a reward function in a straightforward way.

Finally, I am not sure I agree with the conclusions about the efficacy of the approach. I might say it works in the sense that it is not worse than binary/ternary feedback, but that feels like a bit of a strawman argument.

**Summary Of Recommendation:**

Hopefully it is clear from my comments below that I am concerned about the approach and whether this results in a significant enough contribution to the robotics community. There are no major technical flaws that I can see, and like I said above, I think the paper is written well enough. I encourage the authors to think about the fact that (as they state) these problems are multi-objective, and have many qualitatively different classes of solutions.

---

> ### Author Response · Authors · 2021-08-26
> **Response to Reviewer cvuG**
>
> We thank the reviewer for their constructive comments. We are surprised that the reviewer thinks the paper is not very relevant to the robotics community, which we respectfully disagree with. We would like to argue that learning from human feedback is an important component of robot learning. Human feedback can be provided through many different modalities including demonstrations, observations, language instructions, physical corrections, and pairwise comparisons. However, each of these mediums of providing human feedback have their limitations. For example, demonstrations are often difficult to collect on robots with high degrees of freedom; observations require finding the right embedding space and distance metric to be useful; language instructions are often goal-directed and not very informative about the detailed robot trajectories; corrections require physical interaction with the robot. Even though comparisons also have their limitations---specifically every comparison provides at most one bit of information---we strongly believe that pairwise comparisons are a useful, robust to noise, and intuitive medium for acquiring feedback from humans. This is already evident by the large body of robotics work in this area [1,2,3,4,5,6,7,8,13,15,26], and researchers are still working on developing efficient preference-based learning methods in robotics: as an example of a concurrent work, see Lee et al.’s "PEBBLE: Feedback-Efficient Interactive Reinforcement Learning via Relabeling Experience and Unsupervised Pre-training.". Below, we respond to the concerns raised by the reviewer and describe the changes we made thanks to their comments. We made the changes in the paper with blue ink for the convenience of the reviewer.
>
> **[3+ TRAJECTORIES FOR SCALE FEEDBACK]**
>
> When using choice feedback, it is more beneficial to present the users with more than 2 trajectories, as long as the user is not overwhelmed by the number of options. [4] developed the mathematical model for this. From a technical perspective, we can leverage this model for the scale feedback and provide more than 2 trajectories. However, it can create too many practical issues: scale feedback between 3 trajectories requires users to select a point on a triangular region (as opposed to a point on a line). It is cognitively difficult to make such a nuanced evaluation. Scale feedback between 4 trajectories requires selecting a point inside a tetrahedron, which is very difficult without specialized user interfaces. Beyond 4, it becomes almost impossible to elicit feedback. Thus, we used scale feedback only between 2 trajectories.
>
> **[WHY GAUSSIAN NOISE?]**
>
> We use a zero-mean Gaussian noise, as it is the standard probit model. We cited [3,8] as robotics papers that use this model in the preference-based learning context, but others can be found in recommendation systems (see for example Chang et al.’s “Streaming Recommender Systems”, or Lu et al.’s “Recommender System based on Scarce Information Mining”), behavioral economics (see for example Greene’s “Econometric Analysis”, or Arzhenovskiy et al.’s “Multivariate Probit Model for a priori Assessment of Behavioral Risks in Audit”), etc.
> The stationarity of the noise is a common assumption in reward learning: all the reward learning works we cited above make this assumption. A non-stationary model is possible, but harms data-efficiency (see Basu et al’s "Active learning of reward dynamics from hierarchical queries"), and is beyond the scope of our paper.
> Finally, while the noise is independent from the underlying noiseless response $\psi$, it should be noted the full user response model includes an operator that performs clipping and rounding (Eq. (8)). For example, if $\psi=0.9$, then the noise could make it larger than $1$. For any response larger than $1$ (or smaller than $-1$), this operator brings them down to $1$ (or up to $-1$). This is effectively a decrease in the noise variance. Hence, observing $1$ or $-1$ has indeed less uncertainty than $0$.
>
> **[COLLAPSING TO SINGLE DIMENSION]**
>
> Most RL/IRL techniques collapse the objectives to a single dimension by scalarization: the preferences are encoded with a scalar reward. This is also what we are doing, and using comparisons for reward learning is a standard approach in the literature [1,2,3,4,5,6,7,8,13,15,26]. Learning from comparisons is often preferable over the alternatives, e.g. learning from demonstrations, as they provide a simple interface. And importantly, they work as we showed via simulations and user studies, in addition to the real robot experiments conducted in the cited works.

---

> > ### Author Response · Authors · 2021-08-26
> > **Response to Reviewer cvuG (cont.)**
> >
> > **[EXTENDING THE PROPOSITION TO NOISY HUMANS]**
> >
> > Theoretical results do not extend to the noisy case, because adversarial users and/or tasks can be constructed. Imagine a task where the user’s noiseless response $\psi\in\\{-1,0,1\\}$. With soft-choice feedback, under moderate noise, the true response can often be recovered. However, when we go to scale feedback with $\epsilon=0.5$, the noisy responses $-0.5$ and $0.5$ introduce additional uncertainty, potentially causing performance loss. While this is an edge case, we believe a task/user-specific result, where we assume some true distribution for $\psi$, would not be as general and would need to be carefully justified.
> >
> > **[HOW TO CONCLUDE H4 (old H3)]**
> >
> > We conducted additional experiments (see Fig. 6) that could conclude H4, which hypothesizes the users will prefer scale feedback over the soft choice feedback in terms of the optimized trajectories, but we again did not observe significant differences ($p=.1825$).  We suggest in the updated paper that more complex tasks are required for users to differentiate between the methods. However, such tasks would not be suitable for our user studies, as they would require the participants to respond to many queries (as opposed to only 10 queries per algorithm in the current studies).
> >
> > **[EXPRESSIVENESS METRIC]**
> >
> > The meaning of expressiveness is explained to the users: they rated the following statement: “I was able to express my preferences by answering scale/choice questions.”
> > Previously, H5 included one claim for easiness and one for expressiveness. We now divided this into two hypotheses to avoid confusion about our conclusions.
> > Finally, we agree with the reviewer that expressiveness is less important than the user’s opinion of the interface (and the final performance), but we keep the expressiveness metric for completeness.
> >
> > **[SLIDER BAR ANNOTATIONS]**
> >
> > We assume the reviewer refers to Fig. 1 when they say “3 annotations”. This figure is only for visualization, and our approach allows any finite number of steps. We have updated the paper to avoid this confusion. In user studies, we took $\epsilon=0.1$, i.e., the users were able to select $\\{-1,-0.9,\dots,1\\}$ where $-1$ and $1$ indicate strong preference. The bar was interactively annotated with these numbers as the users moved the slider.
> >
> > **[FIRST PARAGRAPH OF INTRODUCTION]**
> >
> > We realized thanks to the reviewer’s comment that some parts of the paragraph were redundant. We have made this paragraph more concise and updated the paper accordingly.
> >
> > **[TYPOS]**
> >
> > We thank the reviewer for pointing out the typos, we fixed them in the updated paper.

---

> > > ### Comment · Reviewer_cvuG · 2021-08-26
> > > **Second response**
> > >
> > > Regarding “Noisy humans”. I agree about the theoretical results not (easily, at least) extending to noisy humans. That was not my intention or point. Rather, the claim is made in several places that there is a worst-case bound, but I cannot see how this is the worst-case bound if the human cannot be noisy (in the proof). It seems reasonable to think that humans will not be consistent. Anyway, my point is that this claim should come with some type of caveat or qualification; not that you need to prove something for noisy inputs, or to require that humans never have noise. This is more about the writing itself and the perceived boldness of the claim.

---

> > > > ### Author Response · Authors · 2021-08-27
> > > > **Response to the Discussion with Reviewer cvuG (cont.)**
> > > >
> > > > **Noisy Humans:** The bound is a worst-case bound for the noiseless human: even when the human is noiseless, it will take multiple queries to learn their reward function. With an inadequate number of queries, the bound describes how bad our worst estimate might be. We have removed the mention of the theoretical result from the abstract in the updated paper, and we now describe it only with the proposition. We hope this addresses the reviewer’s comment about the boldness of the claim.

---

> > ### Comment · Reviewer_cvuG · 2021-08-26
> > **Initial response to the first set of author comments**
> >
> > We can agree to disagree. :) I should caveat my statement or should have stated it more eloquently. I do not think the work is irrelevant. Rather, learning rewards is a very general problem that can be applied to any agent-type of problem, from video games to recommender systems, to applications I probably don’t even know about. This is actually a kind of compliment about the generality of your problem setting. With all of that said, I am happy to modify my assessment of relevance, and I take your points.
> >
> > Regarding your response to “3+ Trajectories”, this is a nice response. These are reasonable, common sensical assertions that are backed by some science, and perhaps you made this choice more clear in the paper than I gave you credit for. To the extent that you would have space in the paper to add a sentence or two to discuss this, that would be great.
> >
> > Regarding your response about Gaussian noice, I totally get it and totally agree. It was more of a philosophical question than an editorial question. Though out of scope (I am not suggesting you do this), I wonder if learners would do better if the choice of noise models better matched the true underlying distributions (which in your application might indeed be 0-mean Gaussian, but in some cases likely has bias and/or is not even Gaussian, which makes the math more difficult of course).
> >
> > Yes, of course RL agents use scalar rewards, and most (all?) optimal control settings involve a scalar cost function. We are in total agreement there. What I am getting at it s more conceptual (even architectural) question. Let us hypothesize that we could agree that a robotics problem involves three objectives, say, time-to-completion, energy consumption, and comfort. Ultimately a RL agent needs to evaluate or determine its policy based on a scalar reward. But for interpretability, expressiveness, etc, it might be important for *us*, the users (or operators, or whatever stakeholder) to be able to tease these things out. Trajectory X was nice because it worked quickly, but I did not like how quickly the robot would run out of batteries (in my made-up example). This also hints at some notion of causality.
> >
> > At least in theory, would it not be possible to have a richer survey of the user’s preference while still ultimately collapsing what is actually *learned* to a single metric? Sure, the many papers you cite do not do this, but who cares? There are fundamental issues with the opacity of how decisions are made, both by humans and deep-learning-enabled agents, and to me this is a richer topic. You may say that this is out of scope; fine, but you kind of invite this type of discourse with the way the paper is written and conclusions are drawn (which is great, by the way!).
> >
> > Also, because something is “standard in the literature”, does this mean it is the best possible idea? Or even “correct”? I do agree that simply showing something works is a good thing.

---

> > > ### Author Response · Authors · 2021-08-27
> > > **Response to the Discussion with Reviewer cvuG**
> > >
> > > We thank the reviewer for discussing our responses.
> > >
> > > **Relevance:** We agree the reward learning problem shows up in many other domains, and our work can be potentially useful / inspiring to other fields. However, our approach is not directly applicable to recommender systems or any system that aims at making profits. This is because the queries in our framework aim at learning the user, whereas in recommender systems the goal is to minimize cumulative regret, i.e., show queries of high-reward items. There is a subtle, but important, difference between these objectives. We explain it with an analogy to bandit problems:
> > > 1. In our information gain approach, we try to quickly learn the reward function everywhere. This is useful in many robotics problems, e.g., when trying to model other agents’ behaviors based on reward. This is analogous to learning the expected payoff of all arms in a bandit problem.
> > > 2. In our max-regret approach, we are trying to quickly learn the best trajectory/policy as in RL. This is similar to best arm identification in bandits.
> > > 3. On the other hand in recommender systems, the goal is to minimize the cumulative regret, so one should not make queries that are informative but consist of low-quality items. This is similar to standard bandit problems.
> > > Therefore, although the scale feedback can be useful in many other domains, our query generation procedures are more specific to the problems where the goal is to train the system as well as possible, just like the problem of reward learning in robotics.
> > >
> > > **3+ Trajectories:** We are happy to see the reviewer found our response reasonable. We have now updated the paper to include the following statements in the Discussion section: “We proposed scale queries as a way to give nuanced feedback between two trajectories. It is possible to extend them to $n+1$ trajectories, with specialized user interfaces that allow users to select a point from a $n$-simplex instead of a slider bar. Future work should investigate this and if users can still give reliable feedback to these more complex queries.”
> > >
> > > **Gaussian Noise:** It is possible that a better user model may exist. One can even think of tuning a user model after collecting a lot of data from multiple users. Such models are often problem-dependent. As an extreme example: if we ask a person whether they want \\$10 or \\$20, they will always choose \\$20. Even if we make the amounts closer, we will not see noisy responses. On the other hand, when the payoffs are not numerical, users tend to be more noisy. Research in behavioral economics shows, as the reviewer suggested, people can also be biased in their selections when, for example, risk is involved (see Kahneman and Tversky’s “Prospect Theory: An Analysis of Decision under Risk” or any other work on prospect theory and risk/loss aversion). However, learning such models often require lots of data, which would be infeasible in our case, because each data point requires the users to watch two full trajectories of the robot. Therefore, even though the active querying algorithm is independent of the choice of user response model, we chose to use a standard model in the literature: the probit model. We have updated the Discussion section to include the following statement:
> > > “Future work should refine the user model, including interactively learning $\sigma$; or fit a new user model that does not necessarily adopt a Gaussian noise.”
> > >
> > > **Richer Information:** We thank the reviewer for the example they gave, which helped us better understand their point. In fact, [6] studied a richer guidance model, as the reviewer suggested, on top of choice questions. In that work, the user was first asked a choice question: “which of the two trajectories do you prefer?“, and upon their response, the user was asked a follow-up question: “which feature has been the most effective in your selection?”. Fortunately, this model is independent of the question type! This means, practitioners could easily augment scale queries with similar follow-up questions. What’s more, the possible answers to the follow-up questions can be included in the mutual information optimization to better optimize the queries for the full response, which includes both the scale feedback and the response to the follow-up. However, this richer guidance requires either the features or a mapping of them to be interpretable. Moreover, one should carefully think about when to ask the follow-up questions. For example, it is not intuitive if the robot asks the follow-up question after the user gives a scale feedback of 0 (meaning “About Equal”), because “which feature has been the most effective in your ‘About Equal’ answer?” is not easy to answer, and does not always make sense. Considering all these different possibilities (and potentially other forms of follow-up questions), we kept them outside of our work to propose and demonstrate the benefits of scale feedback in isolation from the additional features.

---

### Official Review · Reviewer_Z5iG · 2021-07-24

**Originality:** Good
**Technical Quality:** Very Good
**Clarity Of Presentation:** Excellent
**Impact:** 3

**Recommendation:**

Strong Accept: I recommend accepting the paper and will argue for my recommendation even if other reviewers hold a different opinion.

**Summary:**

This paper proposes to learn from a novel type of human feedback in the form of scale-based preferences, i.e. allow a human teacher to indicate how much they prefer one trajectory query over another using a slider bar. Such form of feedback provides more nuanced information compared to binary/ternary choice-based feedback as used in existing preference-based reward learning methods. The authors formulate the problem of learning from scale feedback under the preference learning framework, develop a probabilistic model to account for noisy user feedback and propose two active learning methods that generate queries to maximize information gain and max regret respectively. Experiment in a simulated driving domain demonstrates the advantage of using scale feedback over choice feedback. A user study with a robot performing a shelving task is also conducted to compare learning from scale-based feedback vs choice-based feedback with human inputs. The user study shows that, while scale feedback enables faster learning than choice feedback, users indicated that choice feedback is easier to use.


**Issues:**

The author should also comment on practical limitations of the method, and potential ways of extending this method to real-time interactive learning systems.

**Reviewer Expertise:**

Very good: Comprehensive knowledge of the area

**Strengths And Weaknesses:**

Learning from human feedback is an important research topic that is relevant to many research areas in robot learning. Finding new modalities that a human could provide feedback enriches the ways users could communicate with robots. This paper proposes a novel type of human feedback that an agent could learn from and demonstrate its advantage over existing approaches. The authors also develop a probabilistic model to account for the noisiness in human feedback. The paper is well-written with clarity and the experimental results clearly shows the proposed method’s advantage over the choice-based baseline.


For studying methods that interact with human users, it is very important to also control/measure the human user’s experience. While the authors did a very good job at asking how the users would rank the easiness of the two approaches, it would be useful to also see some objective measures such as how much time it actually took for the user to answer these preference questions on average.
One other small critique I have about the user study design is that the robot’s trajectories are all pre-computed, which is ok for the purpose of the study but does not apply to real applications. It would be nice if the authors could also comment on how to address this limitation if this method were to be deployed on real systems.


**Summary Of Recommendation:**

I highly recommend the acceptance of this paper. This paper proposes to learn from a novel type of human feedback with active learning algorithms that improves efficiency over existing methods. The writing and presentation of the work is well-executed.

---

> ### Author Response · Authors · 2021-08-26
> **Response to Reviewer Z5iG**
>
> We thank the reviewer for their comments and we are glad to hear they found our paper clear and well-written. We made the changes in the paper with blue ink for the convenience of the reviewer.
>
> **[INTERACTIVE LEARNING OF PREFERENCES]**
>
> Regarding the reviewer’s question about how our method can be extended to real-time interactive learning systems without any pre-computed trajectories, we discuss two different possibilities, one for the information gain optimization and one for the max-regret optimization:
> 1- In our paper, we optimized the queries over a set of pre-computed trajectories, which is often desired to avoid the synthesis of unsafe trajectories. However, alternatives that do not require pre-computed trajectories exist. As in [2], when a simulator of the system is available, the queries can be optimized directly over the sequence of robot actions. With such an approach, for example, our Equation (16) would be an optimization over the actions that lead to the trajectories $P$ and $Q$. Of course, this is a much more difficult optimization problem, because the number of variables is much higher and the variables are often continuous. Availability of the gradient of the trajectory features with respect to the action sequence could ease the optimization, but it is often unrealistic to assume in real systems.
> 2- Another possible approach is when the max-regret optimization is used for query selection. In this approach, the optimization is over the reward parameters. In our current implementation, we then find the trajectories in the precomputed set that lead to the highest reward under those optimized parameters. However, a precomputed set is not needed. As in [5], one can use any planner, e.g. a model predictive controller, to optimize the trajectories with the given reward functions.
>
> We added a brief discussion of these in the Discussion section.
>
> **[RESPONSE TIME]**
>
> We did not report the response time of the users in our user studies. This is because we did not impose any time limit and the studies were completely online due to the pandemic restrictions. Hence, our users could technically take breaks between the questions and we would have no way of observing this. Therefore, the response time data would not be reliable or informative. However on average, the first and the second user studies took 24 and 20 minutes per user, respectively. Given the users provided 4 and 3 different datasets respectively; we can say they were able to train the robot in under 10 minutes.

---

### Meta-Review · Area_Chair_kz9D · 2021-08-14

**Recommendation:** Accept (Poster)
**Confidence:** 4

**Metareview:**

Overall, the reviewers found that the paper addressed an important problem and was clearly written. The proposed method is a reasonable and intuitive solution to the problem and was supported by empirical evidence consisting of a user study.

The paper would be improved by more discussion about the impact of the approach, for example, is using a slider to select preferences the right approach or could there be potentially better solutions? What are limitations of the approach? Furthermore, the approach was evaluated with pre-recorded robotic trajectories, and so discussion of how this might scale to more difficult settings would also help make the impact more clear. Finally, there were some concerns about the user study and the conclusions drawn in the paper, and so further evidence or discussion supporting the claims would improve the paper.

Decision:
Thank you to the authors for taking time to address the reviewer's concerns. Overall, I already found the paper to be interesting and believe the paper will be greatly improved with the new updates and experiments.

---

### Decision · Program_Chairs · 2021-09-13

**Decision:**

Accept (Poster)

**Comment:**

Overall, the reviewers found that the paper addressed an important problem and was clearly written. The proposed method is a reasonable and intuitive solution to the problem and was supported by empirical evidence consisting of a user study.

The paper would be improved by more discussion about the impact of the approach, for example, is using a slider to select preferences the right approach or could there be potentially better solutions? What are limitations of the approach? Furthermore, the approach was evaluated with pre-recorded robotic trajectories, and so discussion of how this might scale to more difficult settings would also help make the impact more clear. Finally, there were some concerns about the user study and the conclusions drawn in the paper, and so further evidence or discussion supporting the claims would improve the paper.

Decision:
Thank you to the authors for taking time to address the reviewer's concerns. Overall, I already found the paper to be interesting and believe the paper will be greatly improved with the new updates and experiments.